# In Vitro Induction of Hypertrophic Chondrocyte Differentiation of Naïve MSCs by Strain

**DOI:** 10.3390/cells14010025

**Published:** 2024-12-30

**Authors:** Thomas Jörimann, Priscilla Füllemann, Anita Jose, Romano Matthys, Esther Wehrle, Martin J. Stoddart, Sophie Verrier

**Affiliations:** 1AO Research Institute Davos, Clavadelerstrasse 8, 7270 Davos, Switzerlandesther.wehrle@aofoundation.org (E.W.); martin.stoddart@aofoundation.org (M.J.S.); 2RISystem AG, 7302 Landquart, Switzerland

**Keywords:** human MSC, mechanical strain, hypertrophic chondrocytes, in vitro, 3D, bioreactor, bone healing

## Abstract

In the context of bone fractures, the influence of the mechanical environment on the healing outcome is widely accepted, while its influence at the cellular level is still poorly understood. This study explores the influence of mechanical load on naïve mesenchymal stem cell (MSC) differentiation, focusing on hypertrophic chondrocyte differentiation. Unlike primary bone healing, which involves the direct differentiation of MSCs into bone-forming cells, endochondral ossification uses an intermediate cartilage template that remodels into bone. A high-throughput uniaxial bioreactor system (StrainBot) was used to apply varying percentages of strain on naïve MSCs encapsulated in GelMa hydrogels. This research shows that cyclic uniaxial compression alone directs naïve MSCs towards a hypertrophic chondrocyte phenotype. This was demonstrated by increased cell volumes and reduced glycosaminoglycan (GAG) production, along with an elevated expression of hypertrophic markers such as MMP13 and Type X collagen. In contrast, Type II collagen, typically associated with resting chondrocytes, was poorly detected under mechanical loading alone conditions. The addition of chondrogenic factor TGFβ1 in the culture medium altered these outcomes. TGFβ1 induced chondrogenic differentiation, as indicated by higher GAG/DNA production and Type II collagen expression, overshadowing the effect of mechanical loading. This suggests that, under mechanical strain, hypertrophic differentiation is hindered by TGFβ1, while chondrogenesis is promoted. Biochemical analyses further confirmed these findings. Mechanical deformation alone led to a larger cell size and a more rounded cell morphology characteristic of hypertrophic chondrocytes, while lower GAG and proteoglycan production was observed. Immunohistology staining corroborated the gene expression data, showing increased Type X collagen with mechanical strain. Overall, this study indicates that mechanical loading alone drives naïve MSCs towards a hypertrophic chondrocyte differentiation path. These insights underscore the critical role of mechanical forces in MSC differentiation and have significant implications for bone healing, regenerative medicine strategies and rehabilitation protocols.

## 1. Introduction

Upon injury, the bone healing process and outcome strongly depend on the initial biomechanical environment of the fracture [1]. Direct (also called intramembranous) bone healing requires absolute fixation stability and a reduced fracture gap size [2,3]. In all other cases (the majority), secondary bone healing occurs, whereby endochondral repair is initiated by motion at the fracture site. Bone tissue regeneration is then the result of successive events including the formation of a hematomata, the recruitment and differentiation of mesenchymal stem cells (MSCs) and the formation of a cartilaginous callus that, upon achievement of adequate stability, will remodel towards bone tissue over time [4].

Since its publication in 1980, the strain theory developed by Prof Perren, describing the role of interfragmentary tissue deformation, has formed the basis of theories related to the determination of the bone healing pathway [5]. According to this principle, the route of the bone healing process is regulated by the percentage of displacement present in a fracture gap related to its size (strain).

The correlation between the mechanical environment and callus formation has since become widely accepted, and, for decades, studies have explored the impact of different aspects of mechanical stability on the healing of fractures [6,7,8]. Previous in vivo studies have shown the impact of the interfragmentary motion amplitude (percentage of strain) or the beneficial effect of cyclic loading on callus formation [7,9,10,11]. Other pre-clinical studies, performed in a rat femoral defect, have shown the influence of the bone fixator stiffness, and therefore its flexibility, on callus formation and bone healing [12,13,14].

For a better monitoring of fracture stability/instability conditions leading to complete bone healing, the effect of the mechanical environment needs to be better understood, not only at the tissue level but also at the cellular level. MSCs are key players in bone formation, maintenance and repair and at different stages of bone healing. MSCs are not only responsive to the biological and mechanical properties of their immediate surrounding environment but also to external mechanical signals converted into intracellular biochemical signals [15,16,17,18]. For many years, the development of multiple bioreactor systems has enabled the study of different mechanical conditions, such as fluid flow/perfusion, shear stress or multiaxial loading, through in vitro studies [19,20,21,22,23,24]. Diverse types of mechanical signals have been found to impact MSC differentiation. For instance, mechanical stretching was shown to promote the osteoblastic differentiation of MSCs [25,26]. Likewise, bone marrow MSCs were shown to develop an osteoblastic phenotype in response to shear stress forces [27]. Cyclic compression is another (if not the main) type of mechanical stimulation that has been shown to promote MSC differentiation towards an osteoblastic lineage both in vivo and in vitro [28,29] and could even replace the usage of classical osteogenic media, since it was shown not to directly involve the Runx2 transcription factor, but increased the autocrine BMP levels [30]. Besides the type of mechanical stimulation, its magnitude, frequency and duration play a significant role. Previous studies applying cyclic compression on MSC-seeded scaffolds (tri-calcium phosphate-based) showed increased expression on osteoblast-related genes such as osteonectin, Type I collagen and ALP [28]. In another study, Michalopoulos and colleagues showed that collagen–alginate MSC-seeded scaffolds subjected to 4 h of 10% cyclic compressive strain daily induced cell differentiation towards the osteogenic lineage, while 15% strain applied in the same regimen promoted a rather chondrogenic differentiation of the cells. In the context of articular cartilage tissue engineering, 10% cyclic compression at 1 Hz for 1 h per day and in combination with shear stress led to an increase in chondrogenic gene expression [31].

Many efforts have focused on defining the role and influence of various types of mechanical loading (e.g., shear, cyclic compression) on skeletal-related cells such as osteocytes, osteoblasts and chondrocytes, often in the presence of cell-type-specific differentiation media (e.g., osteogenic or chondrogenic). However, little is known about the effect of mechanical loading on the hypertrophic chondrocyte differentiation of naïve MSCs, which are fundamental cell types in endochondral ossification. In the present study, using a custom-made uniaxial high-throughput bioreactor, we investigate, in a well-defined and controlled manner, the effect of strain on naïve MSC differentiation in the absence of any exogenous growth factor stimulation, we also explore the possible synergetic effect of chondrogenic transforming growth factor β1 (TGFβ1).

A comprehensive understanding of the impact of mechanical stimulation on MSCs during endochondral bone regeneration is crucial in designing rehabilitation training programs for patients following fracture surgery, for the development of mechanically adapted/smart implant systems and for the design of more effective tissue engineering strategies to improve the outcomes of regenerative medicine approaches and ultimately bone healing.

## 2. Materials and Methods

Minimum essential medium alpha (αMEM), Dulbecco’s modified eagle’s medium, 4.5 g/L glucose (DMEM) and penicillin/streptomycin solution (10,000 U/mL and 10,000 µg/mL, respectively) were from Thermo Fischer (Reinach, Switzerland). Fetal calf serum and ITS premix were from Corning. Non-essential amino acids (NEAA) was from Gibco (Life Technology, Reinach, Switzerland), ascorbic acid and 6-aminocaproic acid (EACA) were from Sigma (Buchs, Switzerland). Trypsin–EDTA was from Life Technologies (Reinach, Switzerland). Human recombinant fibroblast growth factor basic protein (bFGF) and TGFβ1 protein were from Fitzgerald Industries (Bray, WI, Ireland). QUANTI RNeasy Mini Kit was from QIAGEN (Hombrechtikon, Switzerland). Porcine gelatin and methacrylic anhydride were from Sigma-Aldrich (Buchs, Switzerland). Proteinase K and phosphate-buffered saline (PBS) were from Sigma.

### 2.1. Samples Preparation

Cell isolation and expansion: Human MSCs were isolated from bone marrow aspirates obtained with the informed consent of all donors. The Swiss Human Research Act does not apply to research that utilizes anonymized biological material and/or anonymously collected or anonymized health-related data. Therefore, this project did not need to be approved by an ethics committee. Patients’ general consent was obtained, which also covered the anonymization of health-related data and biological material (see Institutional Review Board Statement and Informed Consent Statement).

After standard density gradient separation (Histopaque-1077, Sigma), mononucleated cells were plated in polystyrene cell culture flasks at a density of 3000 cells/cm^2^. Cells were expanded in αMEM containing 10% FCS and 5 ng/mL bFGF at 37 °C in a 5% CO_2_ and 95% humidity-controlled atmosphere. Media were changed every second day. Upon 80% confluence, cells were detached using trypsin–EDTA. Cells from 5 different donors and from passage 2 to 4 were used in the present study (female, 18 to 73 years old, average 53 ± 21.5).

Hydrogel preparation and cell encapsulation: Gelatin–methacryloyl (GelMa) hydrogels were prepared from porcine gelatin (Sigma) as previously described [32]. Ten grams of gelatin were dissolved in 100 mL PBS in a 60 °C water bath, and 1.4 mL of methacrylic anhydride (Sigma) was added dropwise. After 3 h incubation at 50 °C and further dilution in 400 mL of PBS, the solution was dialyzed for 7 days through 12–14 kDa membranes against deionized water. After freeze drying steps and ethylene oxide sterilization, the GelMa lyophilizates were store at −80 °C until further use. Human MSCs (40 × 10^6^ cells/mL) were encapsulated in 16% *w*/*v* GelMa dissolved in PBS. An equal volume of 0.15% (*w*/*v* in PBS) LAP photo-initiator (Gelomics Ltd., Brisbane, Australia) was added to the cell-containing GelMa solution, to reach a final cell density of 20 × 10^6^ cells/mL. Customized silicon molds (Ø 5 × 4 mm) were filled with 78.5 µL of the cell-containing preparation and samples were cured for 8 min using a Luna Crosslinker (Gelomics) under visible light at the intensity of 9 mW/cm^2^.

### 2.2. Bioreactor and Mechanical Deformation

Mechanical deformations of the cell-seeded hydrogels were performed in parallel using two custom-made high-throughput uniaxial bioreactor systems (Figure 1A,B) (StrainBot, RISystemAG, Switzerland) designed to fit in a normal cell culture incubator (37 °C, 5% CO_2_, 95% humidity). A rack of 24 pistons simultaneously applied uniaxial compression to the study samples (Figure 1B). An external monitoring system allowed the programming of different deformation protocols in terms of frequency and time of deformation, continuously or divided into several loading segments (Figure 1C), while specific percentages of strain were defined by the thickness of spacers made of polyaryletheretherketone (PEEK).

The cellularized hydrogels were placed in the wells of 24-well cell culture plates in the presence of either chondro-permissive (CP) medium (high-glucose DMEM, 1% volume/volume NEAA, 1% volume/volume ITS-plus, 50 µg/mL ascorbic acid, 100 nM dexamethasone, 5 µL/mL EACA) or CP+ medium (chondro-permissive containing additional 2 ng/mL TGFβ1). For positive control of chondrogenesis, samples were cultured in the presence of classical chondrogenic medium (C+) consisting of CP medium with an additional 10 ng/mL of TGFβ1. Media were changed twice per week.

Based on pioneer in vivo studies demonstrating enhanced callus formation with increasing strain (0–90%) and optimal recovery periods of 2 h between cycles [10], we applied cyclic deformation protocols of 0%, 10% or 30% strain with 5 s loading and 2 h rest intervals for 14 days in CP or CP+ media (Figure 1D). Six samples of 5 different donors were used for each condition and distributed as specified for the different analyses. Cell viability, gene and protein expression were assessed. Biochemical analysis and mechanical property measurements were also performed.

### 2.3. Cell Viability

Live/dead staining: Cell viability was assessed by live/dead staining at day 1 and day 14. Samples (one per condition and time point for each donor) were cut in halves for better core observation, washed 3 times in PBS and immersed in a 10 µM calcein-AM and 5 µM ethidium homodimer solution (#17783, #46043-1MG-F, both Sigma). After 20 min incubation at 37 °C, and three washes in PBS, the samples were placed in αMEM for confocal microscopy observation using a LSM800 Airyscan confocal microscope (Carl Zeiss, Feldbach, Switzerland). After imaging, the samples were fixed in 4% formaldehyde (Formafix AG, Hittnau, Switzerland) for further histology analysis (see Section 2.5.1).

DNA quantification: DNA was quantified in one sample of each condition and each donor at day 0 and day 14. After overnight digestion in proteinase K (0.5 mg/mL) at 56 °C, 100 µL of the samples were added to 100 µL of Quant-iT™ PicoGreen^®^ fluorescent nucleic acid staining. After 5 min incubation, fluorescence was measured with a TECAN Infinite 200 PRO M Plex reader (excitation 485 nm, emission 535 nm). Lambda DNA (100 μg/mL in TE) was used as a standard.

### 2.4. Gene Expression Analysis

RNA isolation: Two samples per donor and per condition were dedicated to gene expression analysis. On day 1 and day 14 of the experiment, total RNA was extracted in 1 mL of TriReagent^®^ containing 5 µL polyacryl carrier (both Molecular Research Center) using a TissueLyser (RETSCH, Haan, Germany)) set at 25 Hz for 6 min. After further 1-bromo-3-chloropropane (BCP, Sigma) phase separation, the RNA-containing aqueous phase was precipitated and washed three times in cooled 75% ethanol prior to air drying. RNA pellets were dissolved in DEPC water, and the RNA purity and concentration were measured with a NanoDrop One system (Thermo Fisher Scientific). Samples were stored at −80 °C until use.

cDNA synthesis was performed with SuperScript™ VILO™ (Invitrogen^TM^, Thermo-Fisher) (for PCR profiling arrays) or cDNA Synthesis Kit TaqMan^®^ Reverse Transcription reagents (Applied Biosystems, Thermo Fisher) and random hexamer primers (for the PCR validation assays) on 1 µg of RNA.

Real-time RT-PCR was performed using a QuantStudio7 Flex instrument (Applied Biosystems). Mechanosensitive and signal transduction-related gene analysis were performed using Applied Biosystems gene assays. Gene expression profiling was performed using a customized TaqMan^TM^ Array Plate (Applied Biosystems, Thermo Fisher Scientific). The expression of genes of interest was further validated using gene-specific TaqMan^®^ assays or a primer probe set (Microsynth, Balgach, Switzerland) (Appendix A). In all cases, 20 ng/µL TaqMan^®^ reverse transcription-generated cDNA was used. Forty standard amplification cycles were run (10 min at 95 °C, 40 cycles of amplification at 95 °C for 15 s and 60 °C for 1 min). Gene expression in the experimental groups was analyzed using the ΔΔct method. Δct was calculated relative to the corresponding mean ct values of the RPLP0 and OAZ1 housekeeping genes. The ΔΔct values were then calculated relative to the day 0 control (average from 5 donors).

For a better visualization of the full range of gene regulation, the results depicted in the heatmap in Figure 4 represent the log2(fold changes) of the ΔΔct values. In Figure 4, the genes were first classified according to their expression levels in the group CP 10% strain day 14. The gene expression for the other groups (30% CP day 14 and 10% CP+ day 14) was sorted accordingly. The log2 upregulated genes are shown in different grades of red, while blue depicts the genes being downregulated.

The regulation of genes related to cell types of interest in the presence of CP and CP+ media was validated. In this case RT-PCR results were calculated using the ΔΔct related to day 0–0% strain and represented as fold changes (2^−ΔΔct). It should be noted that, in the case of Type II collagen and ACAN, the results are presented as 2^-Δct (gene expression at day 14) since they were not detected at day 0.

### 2.5. Protein Expression

#### 2.5.1. Histology and Immunohistochemistry Staining

Paraffin embedding and sectioning. Following fixation in 4% formalin (Formafix AG) and serial dehydration steps in an ethanol gradient (30 to 70%), samples (1 sample/donor/condition) from day 0 and 14 were embedded in gelatin–agarose using an automatic paraffin carousel (Spin Tissue Processor STP 120-2, Microm, Thermo Fisher), followed by paraffin embedding (Tissue Embedding Center EC 350-1, Microm). Five-micrometer-thin sections were cut using a Paraffin HM 355 S Microtome (Microm).

Immunohistochemistry. Extracellular matrix production was characterized by immunohistochemical staining for Type I collagen (R1038, Origen, 1:400 dilution), Type II collagen (CIICI antibody, Developmental Studies Hybridoma Bank, 1:200 dilution) and Type X collagen (X53, ThermoFisher, 1:100 dilution) antibodies. After deparaffination and rehydration in an ethanol series, the sections were treated with 2 mg/mL hyaluronidase (Sigma) to expose antibody-specific binding sites. After blocking using a horse serum solution (Vector Laboratories, 1:20 in PBS-Tween), the sections were incubated overnight in a moist chamber in the presence of mouse monoclonal primary antibodies at the above-mentioned dilutions. After washing steps in PBS-Tween, the sections were incubated with biotinylated horse anti-mouse secondary antibody (Vectastain ABC kit, Vector Labs), at a dilution of 1:200, for 30 min at room temperature. After a washing step in PBS-Tween, specifically bound antibodies were revealed with the Vectastain Elite ABC kit and DAB (both Vector Labs). Slides were cover-slipped with Prolong Gold Antifade reagent with DAPI (for nuclear counterstaining) (Molecular Probes, Invitrogen, Thermo Fisher).

Negative controls, omitting the first antibody, were performed on sections from the 0% CP+ day 14 group. Microscopic evaluation and digital photographic documentation were performed using a bright-field Olympus BX63 microscope (Olympus). The positive control was performed using human bone and cartilage native tissue samples obtained with donors’ informed consent (femoral head, female, 49 years old).

Safranin O Fast Green staining. After deparaffinization and rehydration, sections were stained with Safranin O Fast Green (Fluka, Germay). The slides were washed with deionized water, stained with Weigert’s hematoxylin for 10 min (Sigma-Aldrich), gently washed under running water for 10 min then stained with Fast Green (0.02% *w*/*v*; Sigma-Aldrich) for 6 min to visualize the collagenous matrix (green). After washing with acetic acid (1% *v/v*; Fluka), the samples were incubated with Safranin O (0.1% *w*/*v*, 15 min; Sigma-Aldrich) to stain proteoglycans (red). The staining was revealed over a 25 s incubation period in a 70% ethanol bath; the sections were subsequently dehydrated with a series of alcohols and were mounted in a coverslip (Eukitt, Sigma-Aldrich).

#### 2.5.2. Biochemistry Analysis (GAG)

The glycosaminoglycan (GAG) content of proteinase-K-digested samples and corresponding conditioned media was quantified using the 1,9-dimethylmethylene blue (DMMB) assay (one sample per donor and per condition). Chondroitin 4-sulfate sodium salt from bovine trachea (Fluka) was used as standard, with the highest standard concentration of 1.25 μg/well. Absorbance at 530 nm was measured using a TECAN microplate reader. The DMMB solution was prepared according to Farndale et al. [33]. GAG contents were normalized to the corresponding DNA content.

### 2.6. Mechanical Testing

The stiffness of the samples was measured at both day 0 and day 14 (1 sample/donor/condition). Compression tests were performed with the Instron 5866 (Instron Co, Norwood, MA, USA) using a 10 N load cell. Test to failure was performed with a rate of 0.2% strain/second compression and 0.01 N preload. The samples were measured within a well plate submerged in PBS. To calculate the Young’s modulus, the slope of the linear region in the stress–strain curve was determined.

### 2.7. Statistical Analysis

All experiments were performed using cells from 5 different donors (n = 5 biological replicates) with two replicates (n = 2 technical replicates). Results were analyzed for statistical differences using the GraphPad Prism (Prism 8, 10.1.2, USA) software. Upon normal distribution of the data, a two-way analysis of variance (ANOVA) including Tukey’s multiple comparison test was applied; in other cases, the Kruskal–Wallis test with Dunn’s correction was used. *p*-values below 0.05 were considered significant.

## 3. Results

### 3.1. Construct Integrity and Cell Viability Is Not Impaired by Application of Strain

Macroscopic pictures of the cellularized hydrogels taken on the day of encapsulation and after 14 days of strain application showed no degradation of the gels (Figure 1D). Cell viability was qualitatively assessed by live/dead staining (Figure 2), in which living cells exhibited green fluorescence while dead cells appeared with red-stained nuclei. The cell density and distribution appeared homogenous between all strain conditions at day 14 and when compared to day 0. Likewise, no significant differences were observed when the experiments were performed in the presence of CP (Figure 2A) or CP+ medium (Figure 2B). DNA quantification (Figure 2C: CP medium and D: CP+ medium) performed at day 0 (pink bar) and at day 14 (blue bars) showed a slight increase (not statistically significant) of DNA content between day 0 and day 14. RT-PCR performed using apoptosis-related genes BAX and CASP3, remaining stable over time, confirmed these data (Appendix A).

### 3.2. Effect of Strain on MSC Gene Expression

To evaluate the impact of mechanical deformation on the cells, we examined the effect of different strain levels on the expression of typical mechano-sensitive genes. The heatmap in Figure 3A shows the gene expression fold changes relative to the day 0 no-strain control, after applying 10% or 30% strain for 14 days. In CP medium, 10% strain induced a stronger upregulation of PIEZO2 and TRPV4 (by 3-fold and 2.5-fold, respectively) compared to other tested genes. TRPV4 expression further increased with 30% strain, reaching a 6-fold increase. When 2 ng/mL TGFβ1 was added to the medium (CP+), PIEZO2, TRPV4 and PIEZO1 were further upregulated (showing 6-fold, 25-fold and 1.5-fold increases, respectively).

Our findings also indicated that both 10% and 30% strain induced the upregulation of TAZ and YAP1 compared to the day 0 control, although adding 2 ng/mL TGFβ1 (CP+) lowered their expression (1.6 versus 2.12 for TAZ and 1 versus 1.57 for YAP). The regulation of mechanosensitive genes was generally less pronounced at 30% strain than at 10%, except for TRPV4 and PIEZO1. With CP+ medium, the gene response shifted: PIEZO2 showed greater upregulation than in CP medium, as did TRPV4 and, to a lesser extent, PIEZO1.

Looking at signaling mechanisms (Figure 3B), WNT8A was upregulated by 10-fold in 10% CP and 10% CP+ conditions. The WNT4 gene expression was below the limit of detection in the condition 10% CP+ day 14. WNT5A was only detected in the TGFβ1-containing medium condition. WNT co-receptor LRP5 was detected in all tested conditions, following the same regulation pattern as RSPO1 and WNT8A when 10 or 30% strain was applied. (See Appendix A for related *p* values). 

To assess the overall impact of the strain percentage on naïve MSCs’ differentiation, we used a TaqMan profiler PCR array comprising 80 genes (including housekeeping) grouped into categories covering different cell differentiation stages and paths (see Appendix A for the gene list, acronyms definitions and ordering references).

The heatmap classification of the genes according to their levels of regulation (Figure 4) indicated that, upon mechanical loading, the hypertrophic chondrocyte-related genes Type X collagen, MMP13 and IHH were among the most upregulated genes at either 10 or 30% strain. The chondrogenic-related genes COMP, BMP4 and HAPLN1 were also upregulated upon mechanical stimulation. Due to the calculation method applied in the heatmap representation of the data (ΔΔct related to day 0), Type II collagen did not appear since it was not detected at the day 0 reference time point (see Section 2.4). Genes typically associated with osteogenesis, such as FGFR2, KDR/VEGFR2 and IBSP, were also strongly affected by strain. When chondrogenic factor TGFβ1 was added to the medium (2 ng/mL, CP+), the further upregulation of Type X and MMP13 gene expression was observed. In addition, chondrogenic related-genes such as Type IX collagen, BMP4 and HAPLN1 were further strongly upregulated, while KDR/VEGFR2, VDR, GDF5 or Type I collagen appeared less expressed.

MSC differentiation under strain was further investigated using single gene expression assays of typical osteogenesis-related (Type I collagen, ALP and Runx2) (Figure 5A), chondrogenesis-related (Type II collagen, Sox9 and ACAN) (Figure 5B) and hypertrophic chondrocyte-related markers (Type X collagen and MMP13) (Figure 5C). The results further indicated the hypertrophic chondrocyte differentiation of naïve MSCs when subjected to mechanical leading. It should be noted that while the expression levels for all tested genes are shown as fold changes (ΔΔct) relative to the day 0 control, Type II collagen and ACAN gene expression (typically first detected at around day 7 during in vitro chondrogenic differentiation of MSCs [34]) is presented as level of gene expression at day 14 (Δct).

In CP medium, the application of strain (both 10 or 30%) induced a significant upregulation of Type I collagen when compared to 0% strain at the same time point (*p* = 0.01 and *p* = 0.03, respectively). In CP+ medium, the differences between groups followed the same trend, with a stronger effect of time-in-culture compared to CP medium. ALP expression appeared stable, independently of the presence or absence of strain, but was slightly downregulated between day 0 and day 14 in CP medium. Conversely, in the presence of TGFβ1 (CP+ medium), a general ALP upregulation was observed independently of the application of strain. Likewise, Runx2 did not appear affected by the percentage of strain applied in CP medium, while TGFβ1 induced a significantly higher expression of Runx2 at day 14 for the loaded samples (10 and 30%) compared to day 0 (*p* = 0.0362 and *p* = 0.0069, respectively). Only 30% strain showed led to significant Runx2 upregulation when compared to 0% at the same time point (*p* = 0.026).

No significant difference in Sox9 expression was observed between loaded groups with or without TGFβ1 (Figure 5B). The Runx2/Sox9 ratio, an indicator of chondrogenic or osteogenic differentiation, was not significantly affected by strain in either of the tested media. Similarly, strain did not affect ACAN gene expression in either CP or CP+ medium. In CP medium, a higher expression of Type II collagen was detected when 10% (*p* = 0.0066) or 30% (*p* = 0.038) of strain was applied, compared to 0% strain at the same time point (day 14). As compared to CP medium, the presence of TGFβ1 (CP+ medium) induced an overall higher expression of Type II collagen in all conditions. However, in the presence of TGFβ1, mechanical loading (10 and 30% strain) resulted in a significantly lower Type II collagen gene expression compared to the no-loading condition (*p* = 0.0104 and *p* = 0.0321 respectively).

As seen in our previously described gene profiler experiment, the single gene analysis confirmed a strong upregulation of Type X collagen and MMP13 hypertrophic-related genes (Figure 5C) in the presence of mechanical loading. Type X collagen was significantly upregulated in 10% (*p* = 0.049) and 30% (*p* = 0.02) strain conditions as compared to 0% strain at the same time point in CP medium and in all conditions compared to day 0 (*p* = 0.001). MMP13 only showed slightly higher expression (not significant) in loading conditions when compared to the unloaded samples at the same time point. Interestingly, when TGFβ1 was added to the medium (CP+), mechanical loading no longer affected Type X collagen expression.

Taken together, these results indicate a preferred hypertrophic chondrocyte differentiation path of naïve MSCs in the presence of uniaxial mechanical deformation alone when compared to chondrogenic or osteogenic differentiation. The addition of TGFβ1, on the other hand, does not seem to foster but to rather overwhelm the effect of strain, as shown by the upregulation of most of the genes in all tested conditions compared to control, particularly Type I collagen, ALP, as expected Type II collagen and ACAN, but also the hypertrophic-related Type X collagen and MMP13.

### 3.3. Extracellular Matrix Synthesis

#### 3.3.1. Immunohistochemistry Staining

Immunostaining for Type I, II and X collagen is shown in Figure 6A,B. Negative controls were obtained by omitting the first antibody reaction, while positive controls were obtained on thin native tissue sections (cartilage and bone femoral head of a female 49-year-old patient). Compared to the negative control (Appendix A), Type I collagen staining appeared positive in all conditions (0, 10 and 30%) with CP medium (Figure 6A). On the other hand, no clear expression of Type II collagen was observed in any of the samples in the absence of TGFβ1, while Type X collagen was detected when the samples were subjected to 10% and 30% strain. In the presence of CP+ medium (Figure 6B), Type II collagen was detected in all conditions. In the absence of strain, a strong signal was observed at the periphery of the samples, where the cells appeared smaller when compared to the cells in the center of the scaffold. Likewise, in the absence of strain, TGFβ1 appeared to strongly increase the production of Type X collagen, mainly at the periphery of the samples, while less staining appeared in the center of the samples. The application of 10 or 30% strain induced higher Type X collagen production in CP+ medium when compared to CP medium. In contrast, Type I collagen seemed to be less detected in CP+ medium compared to CP medium in all conditions, i.e., no strain or 10 or 30% strain. 

#### 3.3.2. Glycosaminoglycans Synthesis

Paraffin-embedded thin sections of the constructs were stained for glycosaminoglycans deposition with Safranin O Fast Green (Figure 7A–F). Constructs cultured in CP medium and in the absence of strain (Figure 7A) did not show any GAG deposition. When 10% or 30% of strain was applied to the samples for 14 days, only very low GAG staining was detected (Figure 7B,C). On the other hand, in the presence of 2 ng/mL TGFβ1 in the medium (CP+), more GAGs were detected, as indicated by the reddish staining of the sections (Figure 7D–F). The chondrogenic differentiation potency of the cells was also tested using chondrogenic medium (C+) containing 10 ng/mL TGFβ1 (Appendix A).

Glycosaminoglycan synthesis was also quantified in both the study hydrogels (blue bars) and corresponding medium (red bars). Total GAG was normalized to the corresponding DNA content. Figure 7G shows a stronger production of GAGs in both TGFβ1-containing conditions (10 ng/mL—C+ positive control and 2 ng/mL CP+ study samples) and in the absence of mechanical stimulation when compared to the day 0 control (first bar) and to the CP medium at day 14, independently of the percentage of strain applied. No difference was observed when 0, 10 or 30% strain was applied.

In line with this, the measurement of the construct’s stiffness at day 0 and day 14 (Figure 7H) showed similar Young’s moduli in 0 or 10% strain conditions when the MSC-seeded hydrogels were cultured in the presence of CP medium. After 14 days of culture in the presence of CP+ medium, the samples showed significantly higher stiffness compared to CP conditions with or without strain. Interestingly, when comparing the hydrogels alone (grey bar) or seeded with cells, a significant drop in stiffness (about 10 kPa) was observed in cell-containing gels as compared to cell-free gels.

It should be noted, while the cells showed comparable sizes in the absence of mechanical loading (Figure 7A,D,I), the cells were significantly larger when the samples were subjected to 10% of strain, with a further increase in the 30% strain condition (Figure 7B,C,I).

## 4. Discussion

Since the introduction of strain theory by Professor Perren in 1980 [5], decades of research has explored the relation between the mechanical instability of a fracture and callus formation in vivo. For a deeper understanding of the specific effect of deformation at the cellular level, in the present in vitro study, we explored the impact of mechanical stimulation alone on the differentiation path of naïve MSCs. The findings of this work suggest a preferred hypertrophic chondrocyte differentiation path of the cells when subjected solely to uniaxial mechanical deformation.

### 4.1. GelMa Enables Mechanical Deformation of Embedded Naïve MSCs

Endochondral ossification is a complex process comprising multiple interconnected phases occurring across environments of varying mechanical stiffness, progressing from an initial soft hematoma through hypertrophic callus formation to ultimately remodeled bone tissue [4]. This study investigated the early-stage differentiation of mesenchymal stem cells into hypertrophic chondrocytes within a post-inflammatory soft tissue microenvironment. The choice of the GelMa hydrogel used in this study was dictated by its adaptable stiffness according to the degree of crosslinking, and was previously shown to enable both MSC differentiation and cell migration [35,36]. In preliminary trials, agarose hydrogels, known to support chondrogenic differentiation [37,38], were found to be too brittle in our hands, and the constructs with and without cells did not reproducibly survive the 14 days of stimulation at 30% strain. In comparison, our pilot data showed the suitable resilience of GelMa hydrogels after 14 days of mechanical deformation, even at the highest percentage of strain (30%), as well as homogeneous cell distribution throughout the hydrogel, enabling the better distribution of the forces applied. This was confirmed in the present study by the regular cell distribution, comparable cell viability and apoptosis-related gene expression in all constructs and among the different strain and media conditions.

The application of strain at the cellular level demonstrated a differential mechanosensitive gene regulation response according to the strain level. Bone tissue is always subjected to changes during development, homeostasis and healing, in which cells sense and respond to mechanical stimuli through a cascade of interconnected molecular signals [39]. In our case, 10% strain in CP medium strongly involved PIEZO2, TRPV4, TAZ and YAP1, while PIEZO1 was only detected in the presence of TGFβ1 in the medium. The addition of TGFβ1 further increased the PIEZO2, TRPV4 and PIEZO1 expression, while a higher percentage of strain did not. PIEZO1 and PIEZO2 are known factors in articular chondrocyte mechano-transduction processes [40,41]. It was also shown that a low level of YAP is associated with chondrocyte differentiation, while its higher expression is associated with the hypertrophy of pre-differentiated chondrocytes [42]. The same study also showed an upregulation of the WNT signaling pathway associated with cell hypertrophy under 10% strain and in the absence of further TGFβ1 stimulation. This agrees with our study in which, in the absence of TGFβ1 (CP) and in the presence of 10 or 30% strain, YAP was increased by a factor of two when compared to the TGFβ1-containing medium (CP+). During bone healing, PIEZO1 was shown to be involved during vascularization and the osteoblastic differentiation of MSCs [43,44,45]. Previously, the role of PIEZO2-TRPV4 was described in chondrocytes’ hypertrophic differentiation in osteoarthritis [46,47]. They are also recognized as molecules sensitive to the biomechanical environment of chondrocytes [48,49]. In addition, while TRPV4 is expressed in most of bone cells, it has also been shown in the hypertrophic cartilage of the growth plates [50,51].

In our study, the addition of TGFβ tended to increase the overall cellular response, partially masking the effect of the mechanical loading. The intracellular pathway gene analysis confirmed these results. R-Spondin1 (RSPO1), which is known to play a role in the transmission of mechanical stimuli and is a key player in the WNT signaling pathway [52,53], was strongly upregulated in the presence of 10% strain, whereas its expression was comparatively lower in the presence of TGFβ1. Previous studies have suggested that the RSPO family activates the WNT/β-cat signaling pathway in chondrocytes, along with a decrease in Type II collagen and Sox9 expression, while promoting a hypertrophic chondrocyte phenotype [54,55]. In our study, not only RSPO but also WNT8A and associated LRP5 gene expression were increased at 10% strain. When TGFβ1 was added to the medium, WNT5A gene was upregulated, along with an increase of WNT8A. WNT5A was previously shown to regulate the cell differentiation fate during developmental stages, promoting chondrogenesis while inhibiting chondrocytes hypertrophy [56,57]. Similarly, in the screening of combined physical and biological stimuli on chondrocytes pre-differentiated MSCs, Lee et al. showed that 40% of strain applied to pre-differentiated chondrocytes promoted the hypertrophic differentiation of the cells in absence of TGFβ, along with the canonical WNT signaling pathway, while the addition of TGFβ rather maintained the chondrogenic phenotype of the cells [42].

### 4.2. Uniaxial Cyclic Deformation Induces the Hypertrophic Differentiation of Naïve MSCs

Generally, our data suggest that uniaxial cyclic loading induced a hypertrophic chondrocyte phenotype of naïve MSCs, as shown by the increased MMP13 and Type X collagen and reduced Type II collagen [58], accompanied by an increased cell volume and reduced GAG. Our initial overall gene expression screening underlined the strong upregulation of MMP13 and Type X collagen, as hypertrophic chondrocyte-related genes, upon deformation [59,60]. COMP, known to be highly sensitive to mechanical loading [61], was also among the most upregulated genes upon mechanical loading in our screening experiment, while Type II collagen was not detected in this set of experiments. BMP4, originally chosen as an osteogenesis-related gene, was also among the most upregulated genes upon mechanical deformation. While initially surprising, BMP4 gene expression was previously detected in different bone healing cells, including hypertrophic chondrocytes at the forming callus [62]. Other genes typically associated with osteogenesis, such as FGFR2, KDR/VEGFR2 and IBSP, were also highly upregulated upon the application of strain. These results agree with observations published by Gerstenfeld and Shapiro in a review article in 1996 [63], in which the authors reported the involvement of genes and proteins typically characterized as osteogenic during the endochondral ossification process of the growth plates.

In vivo hypertrophic chondrocytes have been generally characterized as enlarged cell size and high expression of Type X collagen, Runx2 and MMP13, along with lower expression of Type II collagen and Sox9. Likewise, in pathological conditions such as osteoarthritis (OA), in which articular chondrocytes develop a hypertrophic phenotype, the upregulation of Runx2 and ALP has been detected, along with Type X collagen, MMP13 and IHH and reduced Type II [64,65,66]. While bone growth, endochondral bone healing and chondrocyte hypertrophy during OA are different processes, hypertrophic chondrocytes share some similarities.

In our in vitro setup using naïve MSCs, further validation of key genes’ regulation indicated that Runx2 (typically osteogenic) and Sox9 (typically chondrogenic) transcription factors were not affected by the application of strain but rather by the time in culture. Accordingly, the Runx2/Sox9 ratio, an indicator of the cell differentiation path towards osteogenesis or chondrogenesis [67], did not show a clear preferred differentiation route between these two phenotypes. The upregulation of Type I collagen in response to 10% or 30% strain was also confirmed in this set of experiments, while the absolute quantification of Type II collagen gene expression at day 14 remained low, but it showed a factor of 10 up-regulation under strain compared to no strain. The Sox9 and ACAN expression levels remained stable among all conditions and over time. The upregulation of the hypertrophic-related genes Type X collagen and MMP13 [59,60,65] at day 14 under strain conditions was confirmed, further suggesting the MSCs’ differentiation towards hypertrophic chondrocytes upon strain.

In the context of articular cartilage tissue engineering, MSCs have long been considered as a promising source of cells due to their in vitro capacity to develop a chondrocyte phenotype, characterized by high Type II collagen expression, when cultured at a high cell density (pellets) and in the presence of chondrogenic factors such as TGFβ [68,69]. However, instability in the chondrocyte phenotype has often been reported, resulting in further hypertrophic differentiation of the cells. Following the in vitro chondrogenic pre-differentiation of MSCs using chondrogenic growth factors, the removal of TGFβ from the medium resulted in the retention of some chondrogenic-specific markers, along with increasing hypertrophy signs, such as an enlarged cell size and the upregulated expression of Type X collagen and MMP13 [70,71,72]. One could thus hypothesize that the presence of TGFβ during in vitro culture not only promotes the chondrogenic differentiation of MSCs, but also inhibits their natural “aging” process towards hypertrophy. In a similar manner, the early work of Ballock et al. [73] in the context of bone growth and development indicated the inhibitory effect of TGFβ1 on the hypertrophic differentiation of growth plate chondrocytes.

### 4.3. Mechanical Strain and TGFβ1 Differentially Regulate Mesenchymal Stem Cell Fate

In several previous studies, the application of mechanical deformation in TGFβ-pre-differentiated chondrocytes from MSCs of human [74], bovine [38,75], or swine sources [76] has shown a positive effect of cyclic deformation on glycosaminoglycan production and collagen synthesis. In addition, the importance of shear stress, allowing for TGFβ activation, has been underlined to promote MSCs’ chondrogenic differentiation under mechanical loading [77]. It was also shown that mechanical stimulation decreases hypertrophy marker expression in the TGFβ-induced chondrogenic differentiation of MSCs [78,79,80], while, when withdrawing TGFβ signaling, the cells developed a more hypertrophic chondrocyte phenotype. Likewise, the withdrawal of TGFβ1 treatment of pre-differentiated chondrocytes led to the further hypertrophic differentiation of MSCs in a pellet culture [79]. In our study, all data indicated that uniaxial stimulation induced the hypertrophic chondrocyte differentiation of naïve MSCs (no TGFβ pre-treatment), while the introduction of 2 ng/mL of active TGFβ1 in the culture medium (CP+) induced an increase in Type II collagen and ACAN, accompanied by the loss of the influence of mechanical deformation (strain) on gene expression.

The immunohistology staining performed for Type I, Type II and Type X collagen further confirmed our previously discussed gene expression results regarding the effect of strain only (no growth factors) on the naïve MSCs’ differentiation fate. Compared to the negative control (lacking the primary antibody), Type I collagen was detected in all conditions, with stronger matrix staining when the cells were exposed to 30% strain. On the other hand, only very low Type II collagen content was detected in all CP media and in all strain conditions. Interestingly, in the context of OA, and using a mutant mouse model, Lian et al. showed an accelerated chondrocyte hypertrophy upon Type II collagen loss [66], suggesting the active role of this protein in preventing the hypertrophy process.

On the other hand, and as expected, the presence of active TGFβ1 in the medium induced strong Type II collagen expression. Of note, the ring distribution of Type II, Type X and proteoglycan deposition was observed when the samples were cultured in the presence of CP+ medium 2 ng/mL TGFβ1. Oppositely, when the samples were cultured in the absence of TGFβ1, no ring effect was observed, and increased expression of Type I and Type X collagen was observed at 10 and 30% strain, along with increased cell sizes. This border effect has been observed in other studies, and attributed to a lack of nutrient diffusion in the center of the construct [21,77,81,82]. In our case, since this border affect was only seen in the case of the chondrogenic medium, we suspect that the increased GAG production accompanied by the increased stiffness of the samples would, in parallel, decrease their permeability to the nutrients and/or oxygen known (for the latter) to support chondrogenic differentiation [83,84,85].

Our study confirmed that culturing naïve MSC-containing samples in CP+ medium, with or without strain, led to increased GAG synthesis compared to CP medium, as detected by both Safranin O staining and GAG quantification. This increase was associated with greater sample stiffness and a smaller cell size compared to samples subjected to strain without TGFβ1. As anticipated, the GAG content was lower in the absence of active TGFβ (CP medium) compared to the positive control (C+ medium). Additionally, under mechanical deformation, the results correlated with a rounder cell morphology and significantly larger cell size, consistent with hypertrophic chondrocyte differentiation [60,86,87,88]. In contrast, in the study of Bian et al. [78], when MSCs where pre-differentiated using TGFβ3 treatment for 2 weeks, mechanical stimulation appeared to inhibit the hypertrophic differentiation of the pre-differentiated chondrocyte MSCs, while supporting GAG production and Type II collagen expression, confirming the hypertrophy-inhibitory effect of TGFβ1 and -3 [71].

Our data suggest that the application of strain alone induces a hypertrophic chondrocyte phenotype of naive MSCs. The presence of TGFβ1 in the medium was shown to further increase the overall gene expression, including chondrocyte- and hypertrophic chondrocyte-related genes, in all conditions, while increasing the proteo- and glycosaminoglycan production, showing a smaller cell size and superseding the effect of the mechanical stimulation. Overall, these data suggest that strain alone promotes the hypertrophic chondrocyte differentiation of naïve MSCs, while the presence of TGFβ1 predominantly enhances the differentiation of the cells towards chondrocytes over the effect of strain, suggesting that strain and TGFβ1 do not work in a synergetic way but rather involve parallel differentiation paths [78,79,80].

A clear limitation of our study is that we focused our analysis on day 14 of stimulation, which may have caused us to bypass the intermediate stages of cell differentiation, such as the typical chondrogenesis step that usually occurs within 7 days of culture in the presence of TGFβ1, including Type II collagen or GAG deposition [34]. Moreover, naïve MSC populations are not “pure” cell populations; therefore, cells at different stages may react differently, and our results depict an average response. Finally, only cells from female donors were included in this study.

## 5. Conclusions

Despite the limitations listed above, our system robustly showed that deformation alone induced naïve MSCs’ differentiation towards a hypertrophic chondrocyte-like phenotype in the absence of any pre-differentiation treatment or added exogenous growth factors. Specifically, 10% strain is sufficient to initiate this differentiation, while 30% strain did not prove to be of additional benefit, even though the cells were larger. We also confirm the divergence of the differentiation signaling pathway in the presence or absence of TGFβ and under mechanical deformation. We believe that this information is beneficial to the overall understanding of cell biology during endochondral bone healing, would be of great use in fine-tuning fracture treatment approaches and the optimization of implant flexibility for the treatment of bone fractures.

## Figures and Tables

**Figure 1 cells-14-00025-f001:**
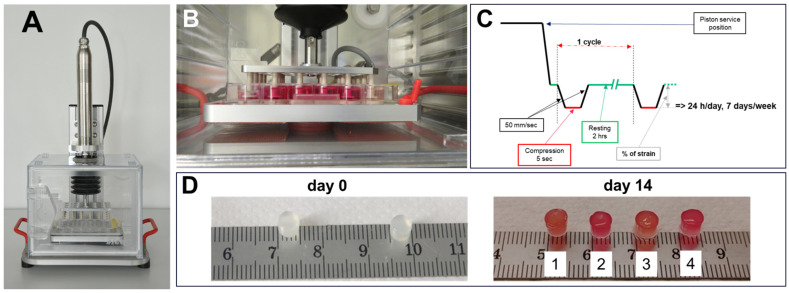
Experimental setup. (**A**) Picture of the StrainBot bio-reactor systems. (**B**) The bioreactor is placed in a cell culture incubator. Samples are placed in each well of a 24-well plate and subjected to mechanical loading for 14 days. (**C**) Illustration of the mechanical loading protocol. Service position corresponds to the position of the piston twice a week for media changes. Green segments represent the rest position of the piston. (**D**) Macroscopic images of cellularized GelMa hydrogel on the cell embedding day (day 0) or after 14 days of mechanical stimulation at 10 (1, 2) or 30% strain (3, 4).

**Figure 2 cells-14-00025-f002:**
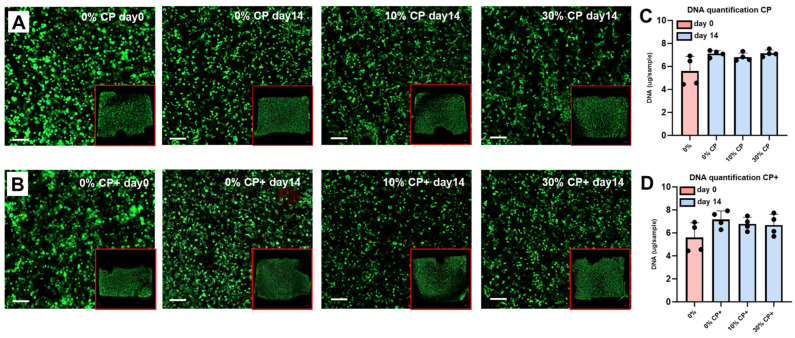
Cell viability of samples subjected to 0%, 10% or 30% strain in either CP medium (**A**,**C**) or CP+ medium (**B**,**D**). Confocal microscopy images (Carl Zeiss) of live/dead staining (**A**,**B**) at day 0 and after 14 days of stimulation show homogenous viability, density and distribution of the cells over time and in all conditions (green: living cells, red: dead cells (scale bar = 100 µm). DNA quantification (**C**,**D**) at day 0 (red bars) and after 14 days of stimulation (blue bars) in presence of CP or CP+ medium confirmed cell viability.

**Figure 3 cells-14-00025-f003:**
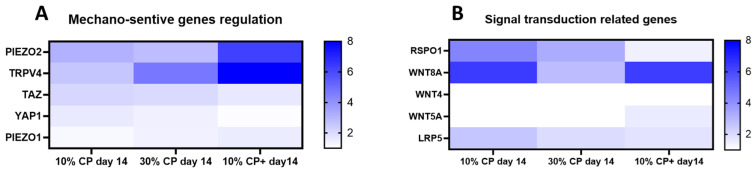
Mechano-sensitive signal regulation. Mechanosensitive (**A**) and signal transduction-related (**B**) genes were analyzed by real-time PCR. Results are presented as gene fold changes related to corresponding genes at day 0 (ΔΔct). Note: TRPV4 fold regulation was higher than the presented range.

**Figure 4 cells-14-00025-f004:**
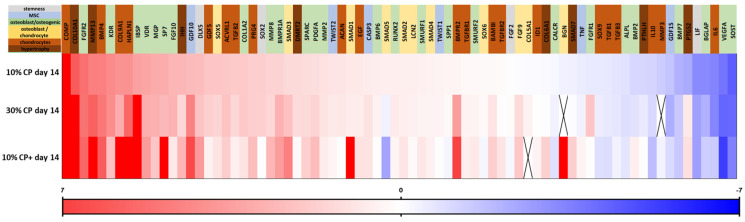
Gene regulation profiling using a TaqMan array comprising a panel of genes grouped by color into different categories. Gene expression levels were calculated by the ΔΔct method related to the day 0 control; heatmap shows the classification of the log2 of the ΔΔct results. The average expression from 5 donors was calculated for each gene and first sorted according to their expression levels in the group 10% CP day 14. Gene expression in the groups 30% CP day 14 and 10% CP+ day 14 was sorted accordingly. The application of strain induced a stronger regulation of hypertrophy (Type X collagen, MMP 13 or IHH) or chondrocyte-related genes (BMP 4, HAPLN 1, GDF5 and KDR). VEGFR 2, IBSP, VDR SP 7 (osteogenic-related) seem to also be sensitive to strain. In the presence of CP+ medium, an overall upregulation of chondrogenic- and hypertrophic-related genes was observed.

**Figure 5 cells-14-00025-f005:**
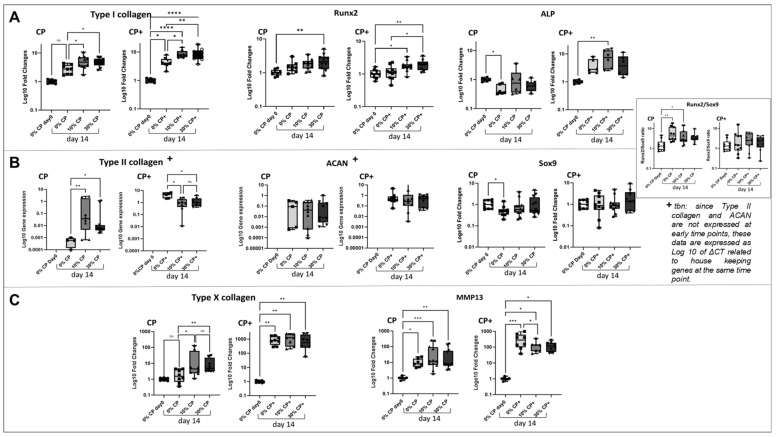
Real-time RT-PCR of osteogenic (**A**), chondrogenic (**B**) and hypertrophy-related (**C**) gene expression in MSCs subjected to different percentages of strain for 14 days, in the presence of chondro-permissive medium (CP) or CP containing 2 ng/mL TGFβ1 (CP+). No significant regulation of RUNX2, ALP, SOX9 or RUNX2/Sox9 ratio was observed according to the percentage of strain in CP or CP+ medium. However, in CP medium, Type I collagen (**A**), Type II collagen* and ACAN* (**B**), MMP13 and Type X collagen (**C**) were upregulated in response to 10% strain compared to 0% or 30% strain. In the presence of CP+ medium, the percentage of strain applied showed less (Type I collagen (**A**) and Type X collagen (**C**)) or inverted (Type 2 collagen, ACAN* (**B**), MMP13 (**C**)) gene regulation effect. + denotes results expressed as log 10 gene expression. * *p* < 0.01, ** *p* < 0.001, *** *p* < 0.0001, **** *p* < 0.00001.

**Figure 6 cells-14-00025-f006:**
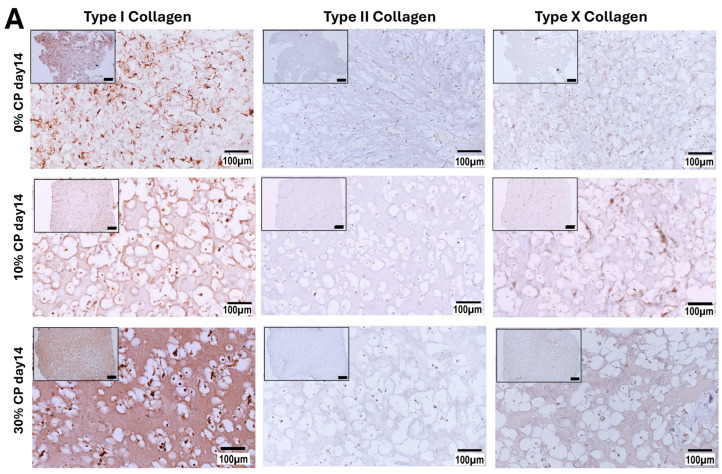
Immunohistology staining of Type I (**left** column), Type II (**middle** column) and Type X collagen (**right** column) of samples subjected to 0, 10 or 30% alone (CP medium, **A**) or in presence of TGFβ1 (CP+ medium, **B**). Type I collagen was detected in all conditions (**A**,**B**) and at different levels. Low level of Type II collagen was detected in all strain conditions and in absence of TGFβ1 (A middle column). Type X collagen staining appeared stronger when 10% strain was applied, increasing under 30% of strain. In presence of TGFβ1, CP+ medium (**B**), Type II and Type X collagen appeared higher expressed compared to CP medium. Type I collagen mainly appeared differently distributed in 30% CP+ medium compared to 30% CP medium. (Scale bar overview pictures = 500 µm, scale bar detail pictures = 100 µm).

**Figure 7 cells-14-00025-f007:**
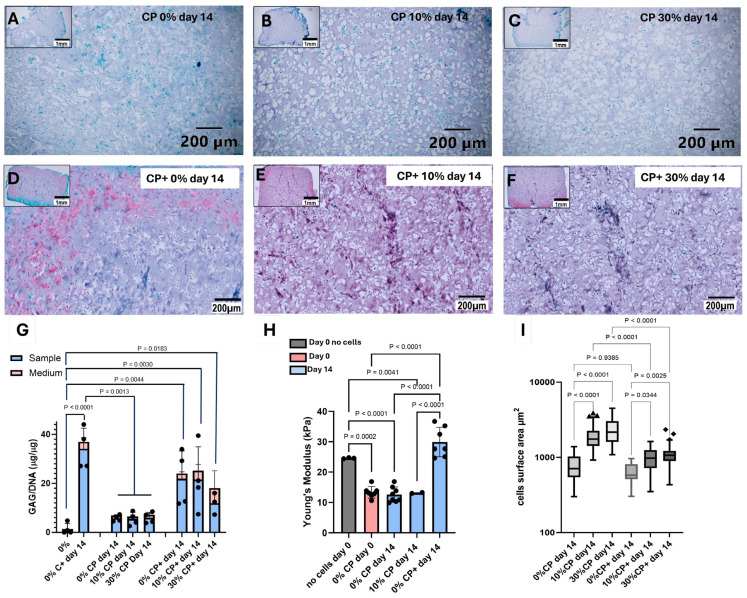
Safranin O Fast Green histology staining performed on samples subjected to 0, 10 or 30% strain in presence of CP medium (**A**–**C**) or CP+ medium (**D**–**F**). (**G**) GAG/DNA quantification showing more GAG production in presence of C+ medium or CP+ medium compared to CP medium. GAG production is more sensitive to the presence of TGFβ1 in the medium than to the percentage of strain. (**H**) Sample stiffness is not sensitive to strain application, but to the presence of TGFβ1. (**I**) Measurements of the cell surfaces in the different tested conditions showed a significantly larger cell size in the presence of 10 and 30% strain compared to no strain in both medium conditions. Despite the application of strain, cells in CP+ medium appeared smaller than in CP medium at the same percentage of strain (scale bar overview pictures (**A**–**F**) = 1mm, scale bar detail pictures (**A**–**F**) = 200 µm).

## Data Availability

Data can be shared individually upon reasonable request addressed to the corresponding author.

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
