# Peer review of "In Vitro Induction of Hypertrophic Chondrocyte Differentiation of Naïve MSCs by Strain"

_cells, 2024, doi:10.3390/cells14010025_

Round 1

Reviewer 1 Report

Comments and Suggestions for Authors

The study of MSC response to mechanical loading has implications for not only endochondral bone formation by endogenous chondroprogenitor cells, but also the tissue engineering of hypertrophic cartilage grafts. Here the authors consider the effects of variable percent uniaxial strain in the absence or presence of TGF stimulus on the differentiation of human bone marrow MSCs. While the chondro-conductive scaffold, the mode of mechanical stimulation, and the chondro-inductive growth factor tested are not individually novel, new insights are made from the combined experimental design. The manuscript is very well written and the results are presented clearly through carefully-constructed figures. The conclusions are consistent with the presented results. I have only a few minor comments:

1. While the observation that uniaxial compression alone promotes hypertrophic differentiation seems novel, the upregulation of hypertrophic markers without pre-deposition of a foundational matrix rich in proteoglycan does not seem fully relevant to the cartilage template priming endochondral bone formation in vivo. Because the study is framed as modeling endogenous MSC differentiation – as opposed to the tissue engineering of hypertrophic cartilage, this potential limitation is not discussed.

2. The authors do not justify the choice of mechanical deformation regimen (strain range and frequency of deformation) or discuss how this choice might have impacted the principal findings.

3. The presentation of gene expression results as heatmaps in Figures 3-4 does not permit the denotation of statistically significant differences among conditions. The results of statical testing could be added within supplemental materials.

4. Section 2.2, line 151: Here it mentions that chondro-permissive medium contains “10 microM ITS-plus”. However, “ITS-plus” contains at least 3 components, so it is not clear how a single molar concentration can be applied to this mixture. Please clarify or revise, as needed.

Author Response

Dear reviewer,

We thank you for taking the time to review our manuscript entitled “In Vitro Induction of Hypertrophic Chondrocyte Differentiation of Naïve MSCs by Strain”. We are grateful for all your comments and suggestions that we addressed to our best in the revised version of manuscript and figures, as well as in our specific responses listed here below. We believe that our work has benefit a lot from your suggestions and would now be ready for publication.

Many thanks,

Kind regards,

Dr. S. Verrier, and on behalf of all my co-authors.

Specific responses:

Rev 1:

The study of MSC response to mechanical loading has implications for not only endochondral bone formation by endogenous chondroprogenitor cells, but also the tissue engineering of hypertrophic cartilage grafts. Here the authors consider the effects of variable percent uniaxial strain in the absence or presence of TGF stimulus on the differentiation of human bone marrow MSCs. While the chondro-conductive scaffold, the mode of mechanical stimulation, and the chondro-inductive growth factor tested are not individually novel, new insights are made from the combined experimental design. The manuscript is very well written and the results are presented clearly through carefully-constructed figures. The conclusions are consistent with the presented results. I have only a few minor comments:

  1. While the observation that uniaxial compression alone promotes hypertrophic differentiation seems novel, the upregulation of hypertrophic markers without pre-deposition of a foundational matrix rich in proteoglycan does not seem fully relevant to the cartilage template priming endochondral bone formation in vivo. Because the study is framed as modeling endogenous MSC differentiation – as opposed to the tissue engineering of hypertrophic cartilage, this potential limitation is not discussed.

We thank the reviewer for his wise comment, with which we fully agree. This point was addressed in the manuscript (page 16, lines 645 to 647), where we mentioned that since we did not analyse any intermediate points (e.g. day 7 when Type 2 collagen and GAG production are usually upregulated), we may have over seen this intermediate step of the process. During pre-hypertrophic chondrocytes stages cells and during further differentiation towards hypertrophic chondrocyte, chondrocytes have been shown to produce enzymes such as collagenase, gelatinase and stromelysin important in the cartilage matrix turnover / degradation in healthy (growth, bone healing) or diseased (AO) conditions [1-3]. In our system possible earlier GAG deposition would have then been already partially degraded at day 14 of our analysis. We have now adapted this point in our manuscript:

Page 16, lines 645 to 647 paragraph

“A clear limitation of our study is that we focussed our analysis at day 14 of stimulation, so we may have missed intermediate stages of cells differentiation, such as the typical chondrogenesis step before hypertrophy, as was previously shown to happened within 7 days of culture in presence of TGFβ1 [34].”  

Has now been changed by:

"A clear limitation of our study is that we focused our analysis on day 14 of stimulation, which may have caused us to miss intermediate stages of cell differentiation, such as the typical chondrogenesis step that usually occurs within 7 days of culture in the presence of TGFβ1, such as Type II collagen or GAG deposition[34]."

  1. The authors do not justify the choice of mechanical deformation regimen (strain range and frequency of deformation) or discuss how this choice might have impacted the principal findings.

We thank the reviewer for bringing our attention to this missing justification of our chosen protocol. The whole concept of this study was based on previous in vivo observations of Hente and Perren (2018). The following sentence has now been included in the manuscript.

Page 4, lines 159 to 162:

“Based on pioneering in vivo studies demonstrating enhanced callus formation with increasing strain (0-90%) and optimal recovery periods of 2 hours between cycles [10], we applied cyclic deformation protocols of 0%, 10%, or 30% strain with 5-second loading and 2-hour rest intervals for 14 days in CP or CP+ media (Figure 1D).”

  1. The presentation of gene expression results as heatmaps in Figures 3-4 does not permit the denotation of statistically significant differences among conditions. The results of statical testing could be added within supplemental materials.

We thank you for pointing out this missing information. In Figure 3, statistical analyses were done using the nonparametric Kruskal-Wallis test with Dunn’s correction, otherwise significance level of the results were proven using the parametric two-way ANOVA test with Tukey’s multiple comparison. To provide a better overview of the genes regulation we decided for a HEATMAP representation of the data. As the reviewer suggested, we have now added a supplemental table 3 (see below) showing the specific level of significance of our results.

Regarding the Figure 4 results, the specific aim was to get an unbiased overview of multiple genes' regulation according to our treatment, and we did not claim significance but rather aimed to provide a regulation overview. Specific genes of interest were thoroughly analysed using classical real time quantitative PCR in Figure 5 where significance levels were added where appropriate.  

  1. Section 2.2, line 151: Here it mentions that chondro-permissive medium contains “10 microM ITS-plus”. However, “ITS-plus” contains at least 3 components, so it is not clear how a single molar concentration can be applied to this mixture. Please clarify or revise, as needed.

Many thanks for pointing out this mistake. We of course meant 1% (volume / volume) of ITS-plus was added. This has now been corrected in the text.

Page 4, lines 152 to 155 now reads as:

“Cellularized hydrogels were placed in the wells of 24-well cell culture plates in the presence of either chondro-permissive (CP) medium (high glucose DMEM, 1% volume / volume NEAA, 1% volume / volume ITS-plus, 50 µg / mL ascorbic acid, 100 nM Dexamethasone, 5 µL/mL EACA), or CP+ medium (chondro-permissive containing additional 2 ng / mL TGFβ1).”

  1. Hall, A.C., The Role of Chondrocyte Morphology and Volume in Controlling Phenotype-Implications for Osteoarthritis, Cartilage Repair, and Cartilage Engineering. Curr Rheumatol Rep, 2019. 21(8): p. 38.DOI: 10.1007/s11926-019-0837-6.
  2. Larson, B.L., et al., Chondrogenic, hypertrophic, and osteochondral differentiation of human mesenchymal stem cells on three-dimensionally woven scaffolds. J Tissue Eng Regen Med, 2019. 13(8): p. 1453-1465.DOI: 10.1002/term.2899.
  3. Sophia Fox, A.J., A. Bedi, and S.A. Rodeo, The basic science of articular cartilage: structure, composition, and function. Sports Health, 2009. 1(6): p. 461-8.DOI: 10.1177/1941738109350438.

Reviewer 2 Report

Comments and Suggestions for Authors

The manuscript by Thomas and coauthors shows mechanical deformation could induce hypertrophic differentiation of MSC encapsulated in GelMa hydrogels. The strain is confirmed by mechanical responsive gene and cellular signaling. The hypertrophic chondrocytes are identified by marker gene, cell size, as well as extracellular matrix components. The evidence is sufficient to support the authors claims and the limitations are clearly listed. This manuscript could be considered for publication in its current form.

Line 293 change figure 1 to 2

Author Response

Dear reviewer,

We thank you for taking the time to review our manuscript entitled “In Vitro Induction of Hypertrophic Chondrocyte Differentiation of Naïve MSCs by Strain”. We are grateful for pointing out our mistake. This has now been changed (cf specific answer here below), and we believe that our manuscript would now be ready for publication.

Many thanks,

Kind regards,

Dr. S. Verrier, and on behalf of all my co-authors.

Specific answer:

Rev 2:

The manuscript by Thomas and coauthors shows mechanical deformation could induce hypertrophic differentiation of MSC encapsulated in GelMa hydrogels. The strain is confirmed by mechanical responsive gene and cellular signaling. The hypertrophic chondrocytes are identified by marker gene, cell size, as well as extracellular matrix components. The evidence is sufficient to support the authors’ claims and the limitations are clearly listed. This manuscript could be considered for publication in its current form.

Line 293 change figure 1 to 2

We thank the reviewer for pointing out this mistake. Figure 1 in line 293 has been changed to Figure 2

Reviewer 3 Report

Comments and Suggestions for Authors

Jörimann et al. used a high-throughput uniaxial bioreactor system to apply varying percentages of strain to hydrogels encapsulating cells. The study demonstrated that deformation alone can induce naïve mesenchymal stem cells to differentiate into a hypertrophic chondrocyte-like phenotype without the need for any pre-differentiation treatment or additional exogenous growth factors. A strain of 10% is sufficient to initiate this differentiation, while 30% strain did not prove to be more beneficial, despite the larger cell size. The results also confirmed the differences in differentiation signaling pathways in the presence or absence of TGFβ and under mechanical deformation.

Major concerns:

1.     Can GelMa be considered a suitable material for simulating the response of human tissues to strain? It is well known that recent studies have demonstrated various mechanical properties of human tissues under external forces, such as stress stiffening and mechanical relaxation. Whether static chemically cross-linked hydrogels like GelMA exhibit these responses and can accurately simulate cellular responses in human tissues under strain is worth considering and discussing.

2.     Different percentages of strain can affect the volume of cells encapsulated in hydrogels. According to the authors' description, cyclic strain promotes the hypertrophic differentiation of MSCs, resulting in an increase in cell volume. This seems counterintuitive. Any data to support such cell volume change? would cyclic compression exert pressure on the cells, causing their volume to decrease? How do the authors understand this?

Minor comment:

1.     The colors in the heatmap of Figure 3 should be chosen for stronger contrast. Additionally, why was the 30% CP + Day 14 group not shown in Figure 3 and Figure 4?

2.     In Figure 5, the data presentation for real-time PCR should be consistent. Additionally, even if there are no statistical differences, it is suggested that this information still be indicated in the figure.

3.     In Figure 6, the font size of the scale bar should be increased. In Figure 7, the font size of the scale bar should be consistent.

Author Response

 Dear Reviewer,

We thank you for taking the time to review our manuscript entitled: “In Vitro Induction of Hypertrophic Chondrocyte Differentiation of Naïve MSCs by Strain”. We have carefully addressed all your feedback into the revised manuscript and figures, with point-by-point responses detailed below. The revisions have enhanced the quality and clarity of our work, and we believe the manuscript is now ready for publication consideration

Kind Regards,

Dr Sophie Verrier, and co-authors

Specific answers:

Rev 3:

Jörimann et al. used a high-throughput uniaxial bioreactor system to apply varying percentages of strain to hydrogels encapsulating cells. The study demonstrated that deformation alone can induce naïve mesenchymal stem cells to differentiate into a hypertrophic chondrocyte-like phenotype without the need for any pre-differentiation treatment or additional exogenous growth factors. A strain of 10% is sufficient to initiate this differentiation, while 30% strain did not prove to be more beneficial, despite the larger cell size. The results also confirmed the differences in differentiation signaling pathways in the presence or absence of TGFβ and under mechanical deformation.

Major concerns:

  1. Can GelMa be considered a suitable material for simulating the response of human tissues to strain? It is well known that recent studies have demonstrated various mechanical properties of human tissues under external forces, such as stress stiffening and mechanical relaxation. Whether static chemically cross-linked hydrogels like GelMA exhibit these responses and can accurately simulate cellular responses in human tissues under strain is worth considering and discussing.

We thank the reviewer for this interesting discussion point. In this particular project, our research hypothesis was that mechanical stimulation alone would trigger differentiation of mesenchymal stem cells (MSC). Our rational in choosing the rather soft GelMa hydrogel (8%), is that we are interested in the early stages of endochondral bone healing where there would be a soft haematoma. Following the resolution of the inflammation phase, when stem cells are invading the bone injury side, and in absence of absolute fixation (driving toward direct ossification), cells are in a rather instable environment and surrounded with a soft fibrin like matrix, thus our choice of GelMa hydrogel.  GelMa is a similarly soft material but has the advantage that it remains stable over time.

Regarding the intrinsic forces applied by the hydrogel to the embedded cells during the curing phase, we believe that those can be neglected since we saw that the cells were still able to migrate in the gels even after curing. In addition, all tested conditions (with or without strain) were done with cells embedded the same hydrogels. Following, this cell-containing hydrogel preparation was poured into silicon mould for curing. Samples were then randomly distributed inf the different groups.

In order to clarify our motivation for GelMa, the following sentences were added to the text:

Pages 13 and 13. Line 484 to 489:

Endochondral ossification is a complex process comprising multiple interconnected phases occurring across environments of varying mechanical stiffness, progressing from an initial soft hematoma through hypertrophic callus formation to ultimately remodelled bone tissue [4]. This study investigates the early-stage differentiation of mesenchymal stem cells into hypertrophic chondrocytes within a post-inflammatory soft tissue microenvironment”

  1. Different percentages of strain can affect the volume of cells encapsulated in hydrogels. According to the authors' description, cyclic strain promotes the hypertrophic differentiation of MSCs, resulting in an increase in cell volume. This seems counterintuitive. Any data to support such cell volume change? would cyclic compression exert pressure on the cells, causing their volume to decrease? How do the authors understand this?

We thank the reviewer for this interesting comment. During bone growth and indirect fracture healing processes, the influence of mechanical stress has mainly been shown in vivo and proven to impair fracture healing in the case of too low or too high percentage of strain at the fracture site ([1, 2]). At the histological level, endochondral ossification (during bone growth and bone healing) is accompanied with the recruitment of stem cells sensing the environment instability ([3]). At the cellular level, mechanosensitive molecules (PIEZO/TRAPV4) [4]  have been shown to respond to mechanical stimulation, promoting IHH gene expression, a key factor in the hypertrophic differentiation of chondrocytes [5] [6]. Hypertrophic chondrocytes have been characterised by an increased volume of 10 to 20 times [7], increased Type X collagen, increased MMP13 [6]. So in our study, this increased size of the cells, along with Type X collagen, MMP13 up-regulation and decreased GAG production corroborate with our initial hypothesis of hypertrophic chondrocyte differentiation of MSCs upon mechanical stimulation, increased size of the cells was measured on the safranin o fast green staining images and reported in figure 7i. To make it clearer, the legend of the y axes fig Figure 7I has been changed form “surface area in µm2” to “cells surface area in µm2

Minor comment:

  1. The colors in the heatmap of Figure 3 should be chosen for stronger contrast. Additionally, why was the 30% CP + Day 14 group not shown in Figure 3 and Figure 4?

We thank you for this comment. We have now changed the colours from black scale to blue scale. We hope that the results are now more visible. Since the aim of this study originally was to investigate the effect of strain alone on the MSCs differentiation, at this stage of the project we only focussed on studying the effect of 0%, 10% and 30% of strain on the cell differentiation. As a control of chondrogenic differentiation potential of a the cells, and due to space limitations in our bioreactor, we only included one extra group comprising TGFβ1. We chose the 10% strain group as we believe it to be the most relevant.

  1. In Figure 5, the data presentation for real-time PCR should be consistent. Additionally, even if there are no statistical differences, it is suggested that this information still be indicated in the figure.

We thank you for this comment. We understand that all data presented in the same figure should be presented in the same way. However, in the case of Type II collagen and ACAN gene expression, since those genes were not expressed at time point 0 (time point we chose for the calculation of the ΔΔCT values giving information on the regulation of the genes compared to this time point, we have decided to still show these gene expression as fold changes (ΔCT) related to house keeping genes at the same time point.

We decided not to express all the other genes as such, as doing so, we would miss some valuable information regarding their regulation. We have now added the following text in the Figure 5:

 “*tbn: since Type II collagen and ACAN are not expressed at early time points, these data are expressed as Log 10 of ΔCT related to house keeping genes at the same time point

  1. In Figure 6, the font size of the scale bar should be increased. In Figure 7, the font size of the scale bar should be consistent.

Many thanks for this comment. We have now increased the size of the police of character beneath the scale bars, we have also re-organised to panels A and B of the picture allowing its further enlargement. We hope this now makes the full reading of figures 6 and 7 easier.

  1. Hente, R. and S.M. Perren, Mechanical Stimulation of Fracture Healing - Stimulation of Callus by Improved Recovery [Mechanická stimulace hojení zlomenin - stimulace svalku prodloužením fáze zotavení]. Acta Chir Orthop Traumatol Cech, 2018. 85(6): p. 385-391.DOI: .
  2. Perren, S.M. and J. Cordey, The concept of interfragmentary strain, in Current concepts of internal fixation of fractures, H.K. Uhthoff, Editor. 1980, Springer: Berlin, Heidelberg, New York. p. 63-77.
  3. Einhorn, T.A. and L.C. Gerstenfeld, Fracture healing: mechanisms and interventions. Nat Rev Rheumatol, 2015. 11(1): p. 45-54.DOI: 10.1038/nrrheum.2014.164.
  4. Zhang, M., et al., TRPV4 and PIEZO Channels Mediate the Mechanosensing of Chondrocytes to the Biomechanical Microenvironment. Membranes (Basel), 2022. 12(2).DOI: 10.3390/membranes12020237.
  5. Mak, K.K., et al., Indian hedgehog signals independently of PTHrP to promote chondrocyte hypertrophy. Development, 2008. 135(11): p. 1947-56.DOI: 10.1242/dev.018044.
  6. Hallett, S.A., W. Ono, and N. Ono, The hypertrophic chondrocyte: To be or not to be. Histol Histopathol, 2021. 36(10): p. 1021-1036.DOI: 10.14670/HH-18-355.
  7. Hall, A.C., The Role of Chondrocyte Morphology and Volume in Controlling Phenotype-Implications for Osteoarthritis, Cartilage Repair, and Cartilage Engineering. Curr Rheumatol Rep, 2019. 21(8): p. 38.DOI: 10.1007/s11926-019-0837-6.